# The Influence of Alloy Composition and Liquid Phase on Foaming of Al–Si–Mg Alloys

**Francisco García-Moreno [1,2,\*], Laurenz Alexander Radtke [2], Tillmann Robert Neu [1,2], Paul Hans Kamm [1,2], Manuela Klaus [3], Christian Matthias Schlepütz [4] and John Banhart [1,2]**

[1] Institute of Applied Materials, Helmholtz-Zentrum Berlin für Materialien und Energie, Hahn-Meitner-Platz 1, 14109 Berlin, Germany; t.neu@tu-berlin.de (T.R.N.); paul.kamm@helmholtz-berlin.de (P.H.K.); banhart@helmholtz-berlin.de (J.B.)

[2] Institute of Materials Science and Technology, Technische Universität Berlin, Hardenbergstr. 36, 10623 Berlin, Germany; laurenz.rdt@googlemail.com

[3] Department of Microstructure and Residual Stress Analysis, Helmholtz-Zentrum Berlin für Materialien und Energie, Albert-Einstein-Str. 15, 12489 Berlin, Germany; klaus@helmholtz-berlin.de

[4] Swiss Light Source, Paul Scherrer Institute, 5232 Villigen, Switzerland; christian.schlepuetz@psi.ch

**\*** Correspondence: garcia-moreno@helmholtz-berlin.de

**Abstract:** The foaming behaviour of aluminium alloys processed by the powder compaction technique depends crucially on the exact alloy composition. The AlSi8Mg4 alloy has been in use for a decade now, and it has been claimed that this composition lies in an "island of good foaming". We investigated the reasons for this by systematically studying alloys around this composition by varying the Mg and Si content by a few percent. We applied in situ X-ray 2D and 3D imaging experiments combined with a quantitative nucleation number and expansion analysis, X-ray tomography of solid foams to assess the pore structure and pore size distribution, and in situ diffraction experiments to quantify the melt fraction at any moment. We found a correlation between melt fraction and expansion height and verified that the "island of good foaming" actually exists, and foams outside a preferred range for the liquid fraction—just above $T_s$ and between 40%–60%—show a poorer expansion performance than the reference alloy AlSi8Mg4. A very slight increase of Si and decrease of Mg content might further improve this foam.

**Keywords:** metal foam; CALPHAD; liquid fraction, X-ray diffraction; X-ray radioscopy; X-ray tomography; X-ray tomoscopy

## 1. Introduction

Although the unusual properties of aluminium foams promise a wide range of applications [1] and various commercial manufacturers are available, the overall production and selling volumes are still quite low compared to, for example, wrought aluminium products or cellular polymers. One reason may be the still too high costs, but it is certainly also necessary to further improve the potential of metal foams by optimising their structure and properties. Two main production routes are commercially available, namely, the melt route and the powder metallurgical route [2]. Foams made by the latter method are stabilized by the action of metal oxide networks hindering film thinning and rupture, as shown by Körner et al. [3], which is crucial for achieving a homogeneous foam. Beside stabilising the liquid foam, other factors, such as the way of creating and distributing gas bubbles during nucleation and further bubble growth, have to be considered. These requirements have led to optimization strategies and related works of various kinds in the past.

The influence of gas nucleation on different alloys was studied by Rack et al. [4] and Kamm [5]. The influence of the different powder compaction methods was described by Neu [6]. In all these

cases, the alloy composition played an important role [4,7]. The foamability of different aluminium alloy systems, such as AlSi [7–9], AlMg [10], AlSiCu [11], AlSiMg, [10,12–14], AlSiCuZn [15] or AlSiMgCu [13], has been reported. The alloy AlSi8Mg4 (in wt.%) has been found to be specially suited and was therefore patented [16] due to the fact of its good foaming behaviour and other favourable properties such as good corrosion resistance. It is now used commercially by the company Pohltec Metalfoam for the production of aluminium foam sandwiches (AFS) [17]. Helwig et al. [13] reported that the large amount of liquid available at low temperatures plays a major role in obtaining a good foam structure, enhancing gas nucleation, and preventing crack formation and propagation in early stages. The presence of Mg in the alloy also facilitates powder compaction and metallic bonding among powder particles due to the breaking of the aluminium oxide layers [18]. On the downside, the macroscopic expansion of Mg-containing alloys can be possibly compromised by increased surface oxidation as demonstrated by Simancik et al. [10].

There has been the claim from Helwig et al. [16] of a narrow "island of good foaming" for alloy AlSi8Mg4 (where numbers define the amount of the alloying element in weight percent, wt.%) with a width of the alloy composition of ±1 wt.% for both components, however, without giving a precise definition of foam quality. The aim of this work was to verify, disprove or further specify this claim in terms of measured quantities, to investigate the role of the liquid phase and its influence on gas nucleation and foaming behaviour as well as to find the most suitable alloy composition for a favourable combination of expansion and morphology.

## 2. Materials and Methods

### 2.1. Sample Preparation

Different alloy compositions in the vicinity of the reference alloy AlSi8Mg4 were produced by mixing elemental or alloyed powders with the blowing agent $TiH_2$. The latter was heat treated for 3 h at 480 °C to match to the alloy's solidus temperature [19]. The powders used are listed in Table 1. The amount of blowing agent was kept constant at 0.5 wt.% for all compositions. For each composition, a total of 30 g of powders were mixed in a tumbling mixer for 30 min and then filled into a cylindrical steel mould of 36 mm diameter. The powders were first cold compacted for 10 s at a uniaxial pressure of 300 MPa to displace as much air as possible and then, in a second step, hot compacted at 400 °C and 300 MPa for 900 s as described in more detail by Helwig et al. [20]. From the resulting cylindrical tablet, rectangular samples of 10 mm × 10 mm × 4 mm and 4 mm × 4 mm × 2 mm size were prepared with a CNC mill to be used in laboratory and synchrotron X-ray radioscopy experiments, respectively. The larger sample surfaces were always perpendicular to the uniaxial compression direction.

**Table 1.** List of powders used.

| Powder | Provider | Purity | $D_{50}$ (μm) |
|--------|----------|--------|------------|
| Al | Aluminium Powder Company Ltd. | 99.7 | 63.9 |
| Si | Elkem AS | 97.5 | 25.7 |
| AlMg50 | Possehl Erzkontor | - | 63.8 |
| $TiH_2$ | Chemetall GmbH | 98.8 | 14.4 |

### 2.2. Laboratory X-Ray Radioscopy and Tomography

For foaming, samples were placed on top of a resistive heating plate (900 W power) leaving a free path for X-ray observation. A thermocouple inserted into the heating plate on its obverse side combined with a CAL 3300 thermo-controller from CAL Controls, Hertfordshire, UK, allowed for the adjustment of the desired temperature profile. The corresponding temperature in the samples was calibrated by reference measurements with a thermocouple inserted into dummy precursors. To analyse the foaming behaviour of the samples, the temperature of the heating plate was raised to 700 °C (corresponding to 673 °C inside the sample) with an average heating rate of ~16.5 K/s and held

there for 180 s, after which the power was switched off and natural cooling led to solidification and conservation of the resulting foam.

The whole foaming process was observed by in situ laboratory X-ray radioscopy. The system consisted of a micro-focus X-ray source from Hamamatsu, Photonics, Hamamatsu, Japan, with a spot size of 5 μm and a power of 10 W (100 kV and 100 μA). The radiographic images were detected by a flat panel detector, also from Hamamatsu, with a 120 mm × 120 mm large field of view and 2240 × 2368 pixels, each one 50 μm × 50 μm in size. Due to the geometrical magnification of 3.5×, an effective pixel size at the sample site of 14.3 μm was achieved. A quantitative analysis of the projected images allows, among others, a detailed calculation of the area expansion evolution of the foams. The equipment has been explained in more detail elsewhere [21].

Post-solidification tomographic images of the foams based on 1000 projections distributed over an angle of 360° were recorded with the same system just by replacing the heating stage by a rotation stage from Huber, Rimsting, Germany.

### 2.3. Synchrotron X-Ray Radioscopy and Diffraction

Simultaneous energy-dispersive X-ray diffraction (ED-XRD) and radioscopy measurements were performed at the Energy Dispersive Diffraction (EDDI) instrument, Bessy II, Berlin, Germany. A scheme of the setup is shown in Figure 1. A detailed description of the system can be found in the literature [22,23]. Samples of 4 mm × 4 mm × 2 mm size were placed on a steel holder on top of a M-660 rotation stage from Physik Instrumente PI, Karlsruhe, Germany, and heated up to 640 °C at a rate of 3.4 K/s with an infrared (IR) heating lamp of 150 W power. The temperature was controlled by a thermocouple inserted into the steel holder beneath the sample. The samples were rotated at 0.2 Hz during the foaming process for better statistics. The transmitted X-ray image was converted to visible light by a 200 μm thick LuAG:Ce scintillator, and the corresponding image was projected onto a Complementary metal-oxide-semiconductor (CMOS) sensor with an effective pixel size of 2.5 μm (DIMAX, PCO, Kelheim, Germany) by a mirror/lens system. We recorded 200 images per second with an exposure time of 3 ms for each. Full diffraction patterns could be recorded in transmission with a multi-channel Ge-detector, model GL0110, Canberra, Lingolsheim, France, under a fixed angle of $2\theta = 6°$.

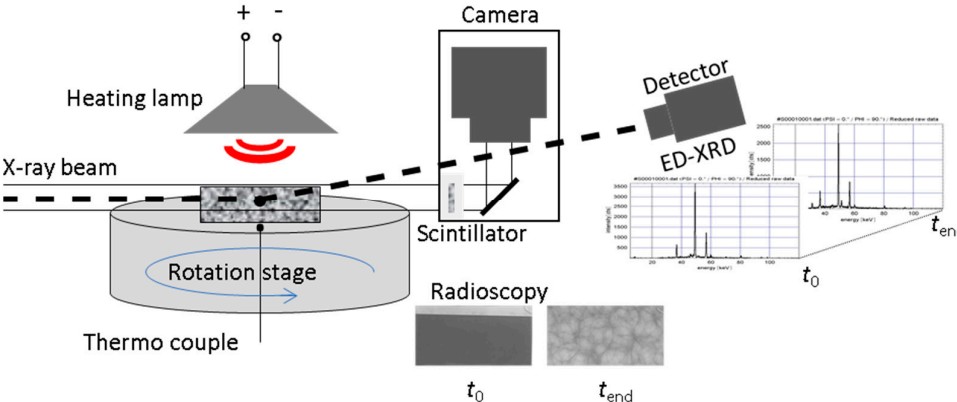

**Figure 1.** Schematic view of the experimental setup for simultaneous X-ray radioscopy and diffraction measurements during foaming of samples at the Energy Dispersive Diffraction (EDDI) beamline, Bessy II, Berlin, Germany.

The ED-XRD patterns were acquired at a speed of 0.4 diffractograms per second throughout the foaming process including melting, foam aging, and solidification. Using the software available at the beamline [22], the solid fraction of the different phases could be calculated from the diffraction patterns by integrating the area of the peaks, each of which corresponded to the volume fraction of the corresponding phase [24]. The liquid fraction in the evolving foam was calculated comparing the

measured peak areas with the areas of the peaks in the solid state. For that purpose, several corrections for the primary beam (wiggler) spectrum, absorption, and detector dead time were applied to the raw data. The individual diffraction lines in the diffraction patterns were fitted by pseudo Voigt functions. Finally, the results were corrected for the changing materials density by taking into account the average X-ray attenuation derived from radioscopy data and Beer–Lambert's attenuation law [25]. As the alloys reduce their density during heating due to the fact of thermal expansion, and this especially during foaming, a density correction was applied. The corresponding density evolution was obtained from the radioscopic images using our own software AXIM [25].

### 2.4. Synchrotron X-Ray Tomoscopy

To further study the foam structure evolution in the liquid state, the recently developed tomoscopy setup at the TOMCAT beamline, PSI, Villigen, Switzerland, was used [26]. The X-ray tomoscopy allowed us to resolve, in 3D, the real foam structure in situ during the foaming procedure, i.e., in the liquid state. Thus, we were able to resolve the gas nucleation stage and the size and shape of the first evolving bubbles, which are a determining factor for the later structural quality of the foam. Samples of 4 mm × 4 mm × 2 mm size were placed in an X-ray transparent boron nitride crucible of 8 mm diameter which was rotated at 10 Hz while acquiring 10 tomograms per second (each covering a 180° angle with 400 projections of 90 μs exposure time, for a total individual scan duration of 0.05 s, separated by 180° rotation without data acquisition) of over a time span of several minutes. The samples were heated up at a rate of ~2 K/s with two IR lasers as described in detail elsewhere [27]. The transmitted X-ray image was converted to visible light by a 150 μm thick LuAG:Ce scintillator and transferred to the GigaFRoST high-speed camera [28] by an optical system with a high numerical aperture and 4 fold magnification [29]. The tomographic projections images were filtered using the propagation-based phase contrast algorithm by Paganin et al. [30] and reconstructed with the gridrec algorithm [31]. The resulting effective pixel size was 2.75 μm.

### 2.5. Calculation of the Liquid Fraction

To predict phase evolution throughout the foaming process, including the fraction of liquid available over the solidification range for the different alloys studied, the commercial software Thermo-Calc 2016a, Thermo-Calc Software AB, Solna, Sweden [32] and a Scheil–Gulliver model [33] was applied. Thermodynamic data were taken from the COST507B database. Thermo-Calc calculates the equilibrium of the phases according to the CALculation of PHAse Diagrams (CALPHAD) method.

## 3. Results

Calculations performed with Thermo-Calc indicate that the amount of liquid fraction should increase with the Si and decrease with the Mg content (see Appendix A and Figure A1). The effects of the variation of the Si and Mg content in the range of ±1% and ±2% around the reference alloy's composition (AlSi8Mg4) on gas nucleation, foam expansion, pore structure, and liquid phase were studied systematically. For all experiments, except the tomoscopy and diffraction analyses, each reported value was the average of three measurements. When showing individual curves or images, a representative sample was selected.

### 3.1. Gas Nucleation and Bubble Evolution

Gas nucleation and bubble growth in the early foaming stage were studied by X-ray tomoscopy. Figure 2 shows representative tomographic slices extracted from tomograms recorded during the stage of bubble nucleation and early growth at three different foaming times and for four representative alloys. At 474 °C, the first gas nuclei can be observed in all samples. They are preferably located at the weakly X-ray attenuating AlMg50 particles. All four alloys exhibited a similar structure at this stage, although slight differences in the number of AlMg50 particles, corresponding to the different Mg contents, can be seen. In the second row, at 570 °C, the difference between the reference alloy AlSi8Mg4 and the others is obvious. While the reference alloy had

already round and smoothly shaped bubbles, the other alloys developed irregularly shaped jagged bubbles. Nevertheless, at 587 °C, the bubble structures of the non-reference alloys seemed to have healed, so that the foam developed an acceptable structure, although with different pore sizes as can be seen later in Section 3.4.

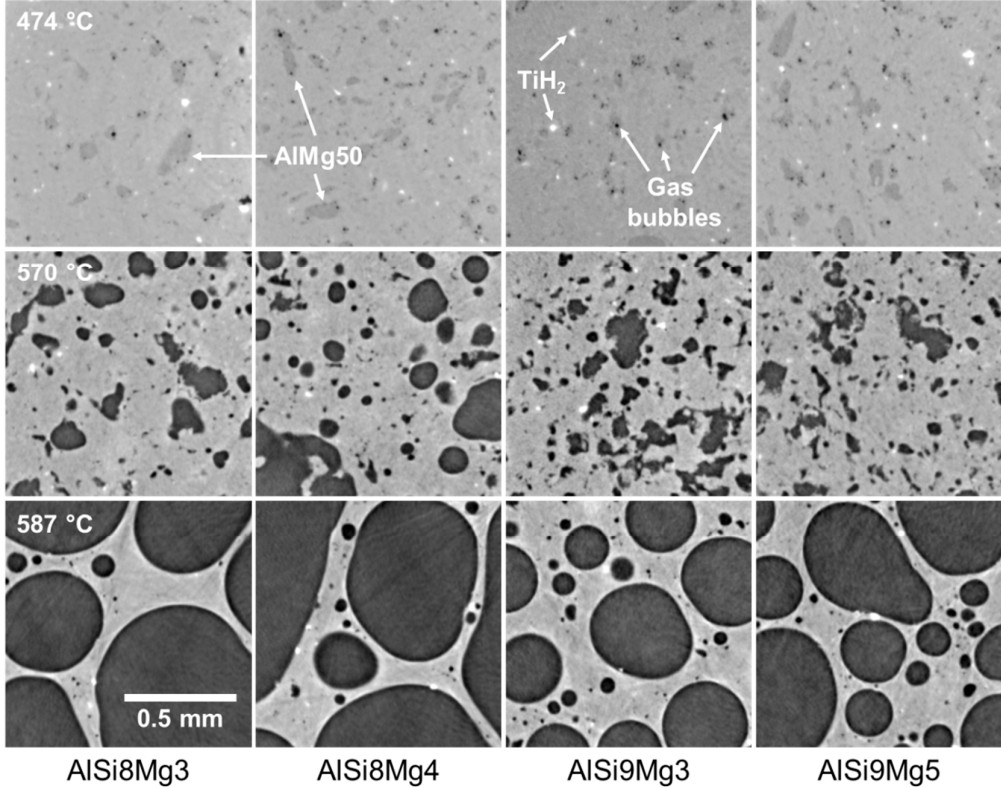

**Figure 2.** Representative tomographic slices extracted from tomograms recorded at three different temperatures for four different alloys. The first row shows the first nucleation stage, the second row the second nucleation stage, and the third row the early bubble growth stage. Gas bubbles, $TiH_2$, and AlMg50 particles are clearly visible in the first nucleation stage and marked with white arrows.

*3.2. Expansion Kinetics*

Figure 3 shows the projected area expansion curves for the selected representative samples of different alloy compositions as derived from laboratory radioscopy data and the corresponding temperature of the heating plate and the calibrated temperature of a dummy sample (a non-foamable piece of a similar aluminium alloy with the same weight and size of a real sample). We observed clear differences in the general course of the area expansion as well as in the maximum or end expansion values of the foams. While alloys AlSi9Mg3 and AlSi7Mg5 reached over 90% of their maximum expansion in ~25 s, AlSi8Mg4 and AlSi9Mg5 needed more than 200 s. Alloy AlSi7Mg5 reached its maximum expansion shortly after the expansion phase and shrunk slowly during the remaining holding time, while the other alloys continued growing slowly until the holding period was over. At the end of the holding period ($t$ = 225 s), the heating power was switched off and after ~40 s (indicated by black arrows in Figure 3), the samples solidified. At this point, a small local minimum in the area expansion can be observed followed by a slight expansion stage. This is known as solidification expansion [34]. The final expansion of the solid foam is the so-called end expansion.

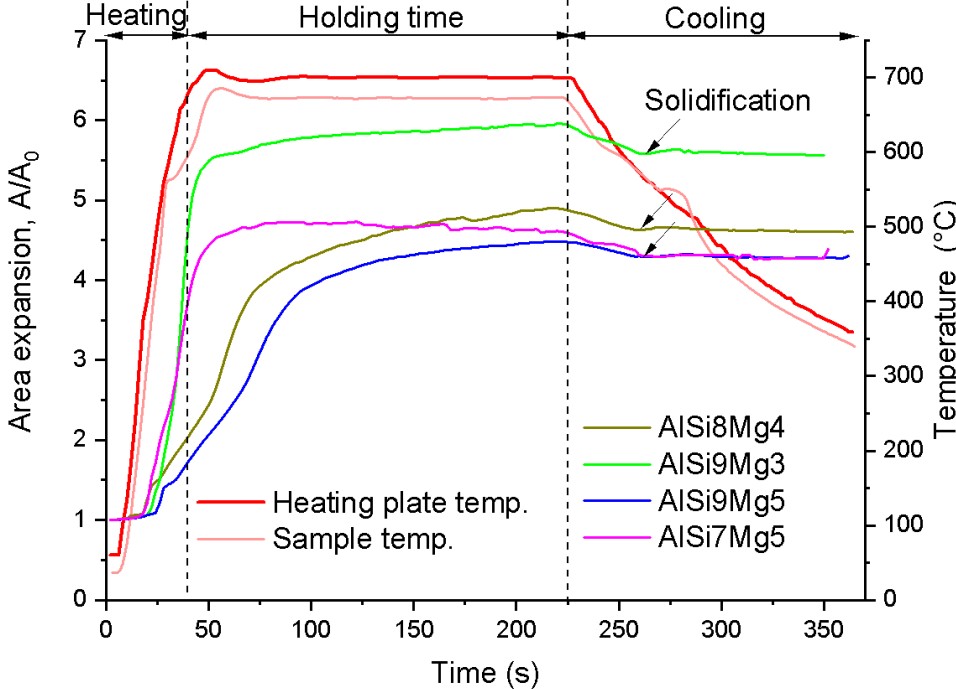

**Figure 3.** Area expansion curves for selected samples of different alloy composition and the common temperature profiles of the heating plate and the sample. Three experimental stages are indicated, namely, heating, holding for 180 s at 700 °C, heating plate temperature, and natural cooling after switching off the power. Black arrows indicate the solidification point in the expansion curve.

### 3.3. Maximum Area Expansion

Mg and Si variations of ±1 wt.% and ±2 wt.% around the reference alloy AlSi8Mg4 were evaluated. The impact of the variation of one element while keeping the other at a constant level on foam area expansion is given in Figure 4. In Figure 4a, we can observe that changes in the Mg content in the range 3–5 wt.% had little impact on expansion, but beyond this, range expansion is reduced. Figure 4b shows a flat maximum of expansion around 9 wt.% Si with a decrease in expansion for lower and higher values. The difference between the maximum and the end expansion expresses the tendency of shrinking (or of collapse in extreme cases) of the foams after solidification. Here, we can see for both variations that the difference was small in all cases.

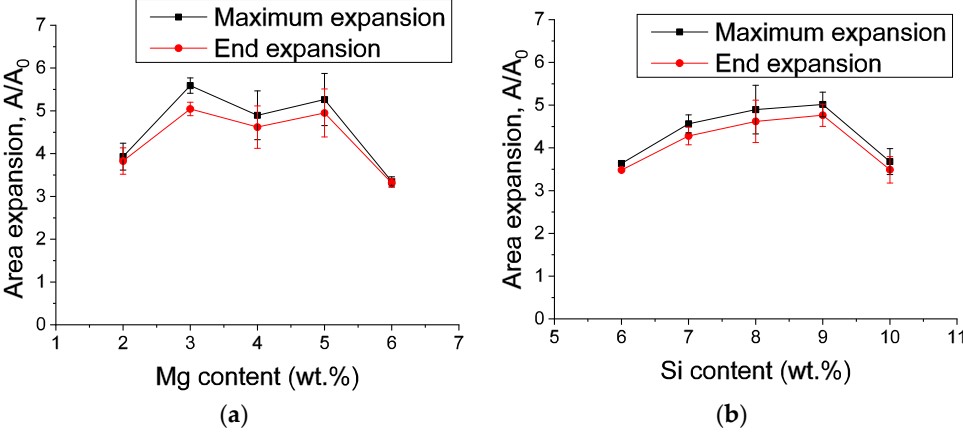

**Figure 4.** Maximum and end foam area expansion given as a function of (**a**) Mg content, with Si fixed at 8 wt.%, and (**b**) Si content, with Mg fixed at 4 wt.%.

In order to also describe simultaneous changes in Si and Mg composition, Figure 5 shows the results of a wider selection of compositions around the reference composition (in blue). Compositions in green show a higher maximum expansion and in red lower ones. The trend of higher expansions seems to be shifted to more Si and less Mg than for the reference alloy.

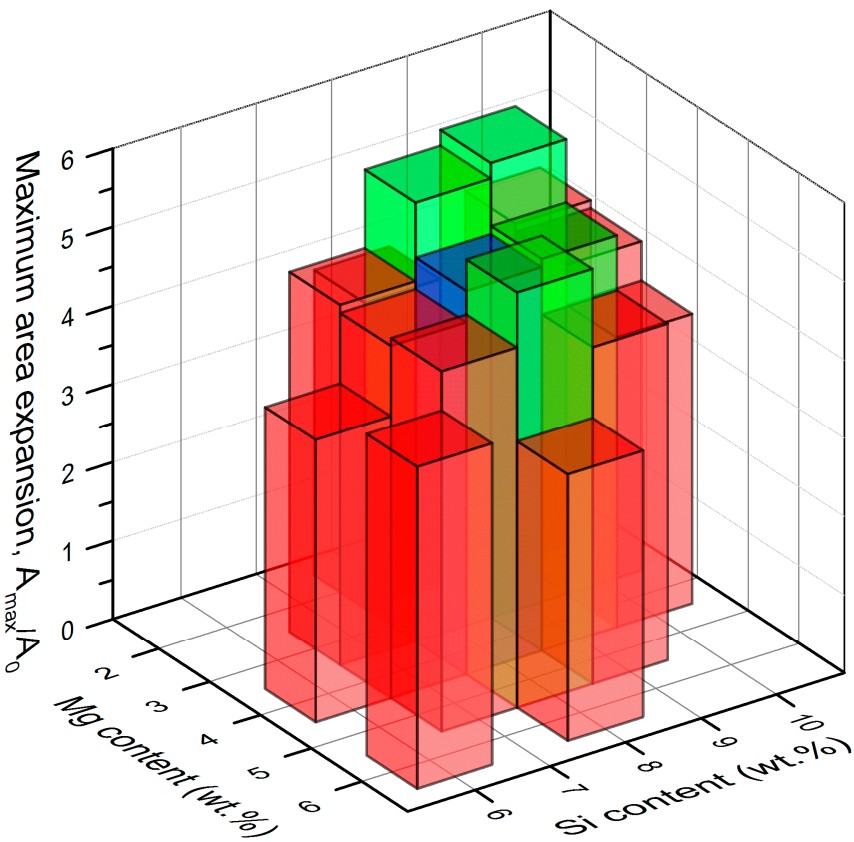

**Figure 5.** Maximum foam area expansion for varying Mg and Si content. Composition in blue is the reference alloy AlSi8Mg4. Compositions in green show a higher maximum expansion and in red lower ones. Only selected alloys around the reference alloy are plotted for a better overview.

*3.4. Morphology*

The evolution of the inner structure of liquid metal foams was elucidated by X-ray radioscopy. Examples of the nucleation stage, maximum expansion, and end expansion for alloys of composition AlSi8Mg$Y$ ($Y$ = 2–6 wt.%) and AlSi$X$Mg4 ($X$ = 6–10 wt.%) can be observed in Figures C1 and C2 of Appendix C. It is possible to observe how cracks induce later large bubbles as well as local coalescence and drainage are induced by gravity at the bottom part of the foams, especially for Si contents >8 wt.%.

The interior of solid foams can be assessed qualitatively from macrographs of sectioned foams (Figure 6). Obviously, minor changes in alloy composition lead to visible changes in pore size. Alloy AlSi7Mg5 (Figure 6a) shows the most irregular structure with a thick outer skin and large non-spherical pores. The reference alloy AlSi8Mg4 and alloy AlSi9Mg5 present the most homogeneous structure, combining a good expansion with spherical pores and a homogeneous pore size distribution. The remaining alloy AlSi9Mg3 still had a good expansion and an acceptable structure but with larger and more polyhedral pores. A ~1 mm thick dense layer can be observed at the bottom of alloys AlSi7Mg5 and AlSi9Mg5 caused by gravity-induced drainage.

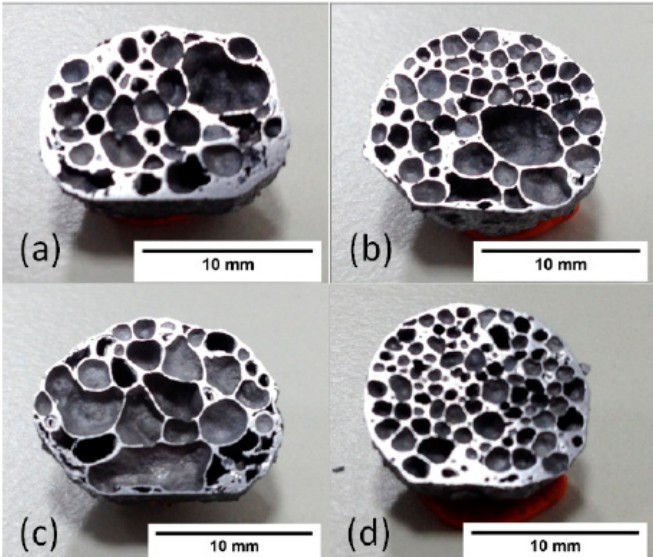

**Figure 6.** Cross-sections showing the pore structure of (**a**) AlSi7Mg5, (**b**) AlSi8Mg4, (**c**) AlSi9Mg3, and (**d**) AlSi9Mg5 after foaming and cutting into two pieces along a plane perpendicular to the sample substrate.

Drawing conclusions from individual foam samples, such as the ones shown in Figure 6, can be treacherous, as their volumes are small and statistical scatter among samples is large. Therefore, additional samples were quantitatively analysed by laboratory tomography, followed by pore segmentation as shown in Figure 7, where different colours indicate separated pores. The volume-weighted pore size distributions can be approximated by Gaussian functions. The mean pore size, sharpness of the distribution in terms of standard deviation, sample volume, and number of pores can be found in Table 2. The trends observed in Figure 6 are confirmed.

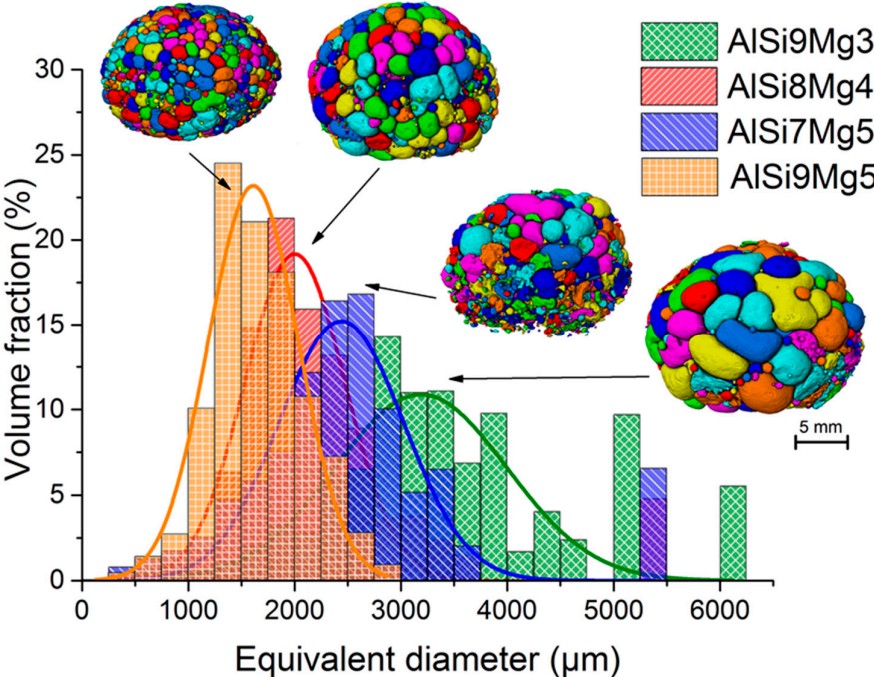

**Figure 7.** Volume-weighted pore size distribution histograms of four selected foams and corresponding Gaussian fits. Insets show rendered images of the segmented pore structures.

**Table 2.** Characteristic values of the foam structure obtained from tomographic analyses.

| Alloy | Mean Pore Size (mm) | $n$ | $V$ (mm³) | $n/V$ (1/mm³) |
|---|---|---|---|---|
| AlSi8Mg4 | 2.00 ± 0.47 | 1189 | 1704 | 0.69 |
| AlSi7Mg5 | 2.44 ± 0.59 | 1086 | 1240 | 0.88 |
| AlSi9Mg3 | 3.19 ± 0.81 | 448 | 2181 | 0.21 |
| AlSi9Mg5 | 1.61 ± 0.42 | 1276 | 1355 | 0.94 |

### 3.5. Liquid Phase Evolution

To evaluate the evolution of the liquid phase during melting and solidification, the foaming process of the alloys was monitored by simultaneous in situ X-ray radioscopy and diffraction analysis. The results were compared with simulations of the solidification process carried out with the CALPHAD method.

Figure 8 depicts density maps of diffracted intensities (grey scale) dispersed in energy (in keV) of the reference alloy AlSi8Mg4 as a function of the time which is correlated with the temperature via the heating curve $T(t)$. Four selected radioscopic images of the evolving metal foam denote its structure and density at different positions. The phases are represented by the energy-dispersive peaks and labelled below these with the corresponding indices. The melting and solidification stadia showing peaks of solid phases mixed with a blurred area from the scattered intensities of the molten fraction are located between dashed blue and red lines (semi-solid phase). The fully molten stage is located between red dashed lines. It can be observed that the $Al_3Mg_2$ phase contained in the original AlMg50 particles disappears after melting and foaming, while the amount of the $Mg_2Si$ phase increases after solidification (see also Figure D1).

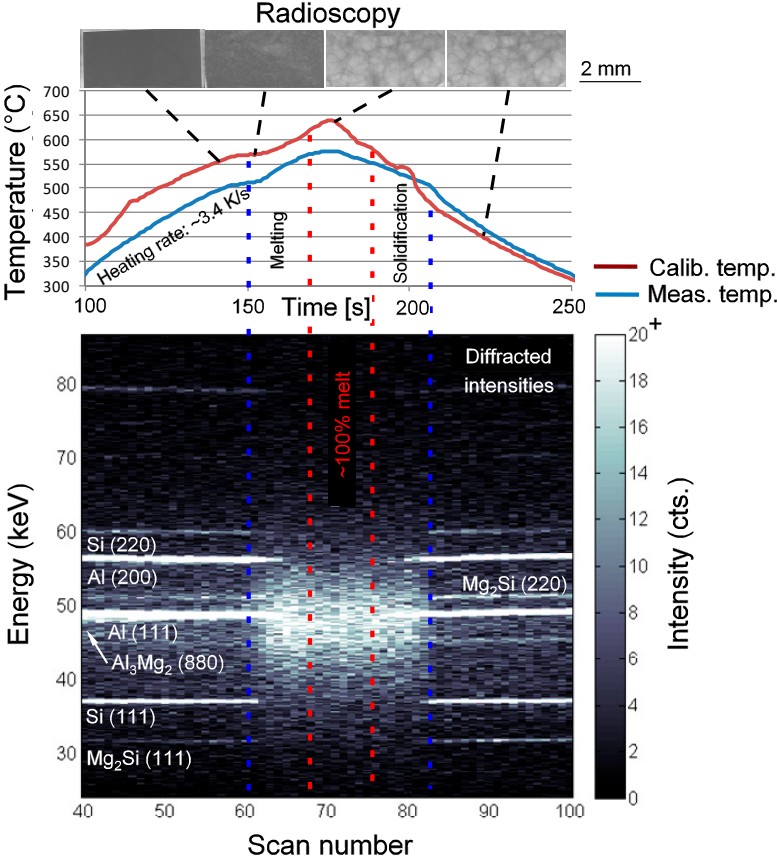

**Figure 8.** Density maps of diffracted intensities (grey scale) dispersed in energy (in keV) versus time correlated to the scan number and to the temperature profile and selected radioscopic images of an evolving AlSi8Mg4 foam.

The evolution of liquid fractions with varying temperature during foaming of four alloys was extracted from the density maps of diffracted intensities and plotted together with the calculated liquid fraction in Figure 9. The black curve represents the relative X-ray absorption of the samples extracted from the integrated intensities of the radiographies which correlates with their relative density via the Beer–Lambert equation [25]. The reference alloy AlSi8Mg4 shows a narrow melting interval starting from 580 °C. It is completely molten at 615 °C. We assumed that the absolute temperature in this case shifted to a higher temperature by ~20 K due to the insufficient sample contact to the heating plate. The AlSi7Mg5 presents the widest melt interval 555–625 °C and had a liquid fraction of 50% at 590 °C. The melting interval of alloy AlSi9Mg3 started at 560 °C and ended at relatively low 600 °C. The liquid fraction was 50% at 570 °C. The AlSi9Mg5 possessed the narrowest melting interval, and for temperatures >580 °C, only 5% solid phase remained.

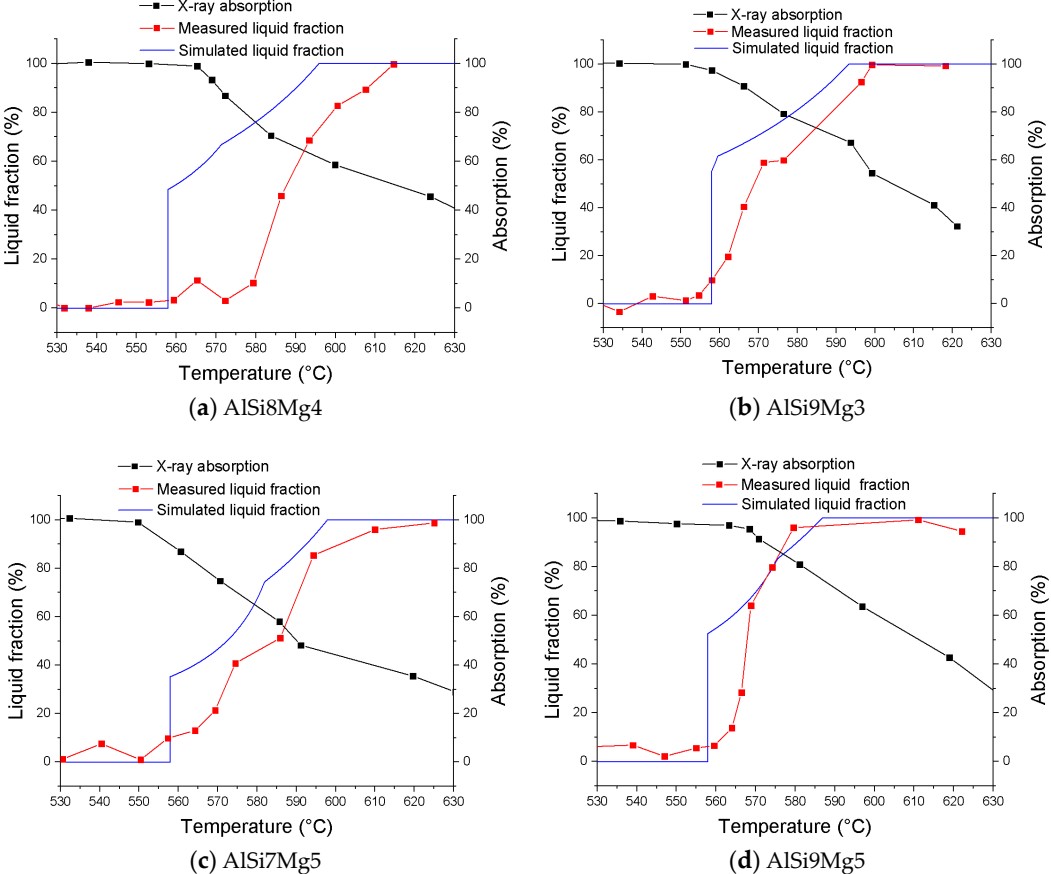

**Figure 9.** Development of the liquid phase over temperature extracted from the density maps of diffracted intensities (in red, see also Figure 8), calculated curves (in blue) and X-ray absorption of the samples related to the absorption of the precursors during foaming of (**a**) AlSi8Mg4, (**b**) AlSi9Mg3, (**c**) AlSi7Mg5, and (**d**) AlSi9Mg5 alloys.

## 4. Discussion

Melting of a foamable precursor is a mandatory step in the production chain of a metal foam following the PM route. It has been claimed that a narrow temperature range in which the liquid phase appears rapidly is beneficial [13]. To study the influence of the liquid phase on foam quality, we varied the alloy composition and found systematic changes. However, variations of composition may also affect other factors influencing foam quality including the quality of powder compaction. For satisfactory foaming, a high density and gas tightness have to be achieved. As shown in previous work on AlSi*X*Mg*Y* precursors, hot compaction at 400 °C and 300 MPa gives rise to good metallic bonding and relative densities above 96% which, in turn, suppresses crack formation in the early

nucleation stages [20] as opposed to, for example, binary AlSi*X* alloys [4], where cracks occur that later lead to large pores and an inhomogeneous pore size distribution [35]. So far, no notable crack formation in the early stage was observed for all alloys studied here as shown in Figure 2, so that strong gas losses through open cracks all the way to the surface can be considered minimal. The increase or decrease in Si content does not have an influence on the density of the precursor, while an increase in Mg has a small positive effect on the compaction density, for example, from 96.1% relative density for AlSi7Mg3 to 97.5% for AlSi7Mg5 [36]. During compaction at 400 °C and 300 MPa, Mg atoms can diffuse into Al and react to form spinel $MgAl_2O_4$, break the $Al_2O_3$ oxide layer [37], and lead to improved metallic bonding and higher densities. On the other hand, this positive effect might be compensated by external oxidation of the entire foam thus hindering free foam expansion [10].

The rate of desorption from the blowing agent $TiH_2$ increases rapidly above 400 °C. An oxidising pre-treatment of the $TiH_2$ powder is known to be beneficial for foaming [19]. Gas nucleation starts already at ~450 °C close to or at AlMg50 particle surfaces, followed by a second nucleation step once the solidus temperature of 558 °C of the AlSi*X*Mg*Y* system has been reached [26]. The first nucleation step may be slightly influenced by differences in the amount of AlMg50 particles, although no significant influence is observed in Figure 2. After reaching the solidus temperature, the role of the amount of liquid fraction becomes visible, as it influences the second nucleation step and the following bubble growth as observed in Figure 2. In alloys like AlSi8Mg4 and AlSi9Mg5, which both have a short melting range, a large number of small pores appear in the initial state of the second nucleation step, leading later to foam structures with smaller pores (see Figures 6, 7, and B1).

The alloy AlSi7Mg5 (Figure 6a) exhibits an irregular pore structure and a thick outer skin caused by early shrinking and collapse as evident from Figure 3. The most homogeneous structures and a good expansion can be found for AlSi8Mg4. The AlSi9Mg5 exhibits a combination of good expansion with spherical pores and a sharper pore size distribution (see Table 2 and Figure 7). Alloy AlSi9Mg3 had the highest expansion of all tested alloys, as shown in Figure 3, but at the cost of an inferior pore structure, namely, larger polyhedral pores (see Figure 6c).

The liquid fractions in the melting and foaming alloys were measured by synchrotron diffraction experiments. An advantage of this method is that the real liquid fraction can be directly measured even if the system is not in thermodynamic equilibrium which is the case here. A disadvantage is that the quantification based on the peak area relies on several corrections as stated previously and is not exact but a reasonable qualitative approximation. Moreover, a direct temperature measurement is difficult because insertion of thermocouples into the foam would have an impact on foaming. Use of calibrated temperatures induces uncertainties.

The in situ diffraction method also allows for an identification of the phases involved. We could observe the disappearance of the $Al_3Mg_2$ phase present in the original precursors contained in the AlMg50 powders after melting and solidification as well as the increase of solid $Mg_2Si$ phase after solidification which was already present in the precursors in lower amounts, indicating its formation already during hot compaction prior to foaming. Solid $Mg_2Si$ exists in the melt as shown by Paes et al. [38]. Pronounced formation of solid $Mg_2Si$ in the early stages of melting can reduce the amount of liquid fraction as was observed in alloy AlSi7Mg5, explaining the early foam stabilisation and the corresponding flat expansion profile in Figure 3, although the presence of $Mg_2Si$ in the liquid stage could not be resolved by diffraction in Figure 8 due to the small quantities present and the strong scattering effect of the liquid. In all cases, the Si peaks disappeared abruptly between 560–580 °C, in the proximity to the eutectic temperature of AlSi, $T_e$ = 577 °C. For AlSi8Mg4, the liquid fraction reached 100% at 615 °C instead of at the liquidus temperature, $T_L$ = 595 °C, indicating that there was a shift of 20 K, probably caused by artefacts of the temperature measurement.

The highest expansions achieved corresponded to theoretical liquid fractions of 40–60% just after passing the solidus temperature. We assumed that the more liquid phase was formed in the early stages, the less gas losses due to the cracks take place, leading to better expansions due to the more effective use of the available gas and a more simultaneous nucleation. On the other hand, if too much liquid phase exists, the lack of solid particles and the corresponding low viscosity deteriorate foam

stability, leading to coalescence and collapse and indirectly to a lower expansion. As a summary of the expansion performance of the alloys studied, Figure 5 clearly illustrates that some compositions (in green) showed higher maximum expansions, while others (in red) gave rise to lower expansions than the reference alloy (in blue). There was a tendency to higher expansions towards slightly higher Si and lower Mg contents than in the reference alloy. However, with 10 wt.% Si, only lower expansions could be achieved. It is known that Si lowers the viscosity of melts, whereas Mg increases it [39]. In Mg containing alloys, $Mg_2Si$ particles are present in the melt [38], although they could not be detected in the in situ diffraction experiments, most likely due to the strong scattering by the molten phase. Low Mg contents lead to large pores due to the less stabilising oxides [3], fewer $Mg_2Si$ particles, and lower viscosities as can be observed in Figures 6 and 7.

As a prediction and comparison, the amount of melt/liquid phase during melting and solidification was calculated using the CALPHAD method. The results obtained show the right tendency, as can be observed in Figure 9, but do not match the measurements exactly, because the method considers the amount of liquid phase during solidification of the already formed alloy. During melting of the foamable samples, the nominal composition and the corresponding phases in the thermodynamic equilibrium have still not formed, since the compacted samples mainly contain elemental powders (see Table 1), and diffusion during the short heating time does not allow for a full dissolution of the components. As shown in Figure 9, we can conclude that alloys with high Si contents show a narrower melting interval and lower temperatures for complete melting, while the influence of Mg is not clear, possibly due to the effect of a forming $Mg_2Si$ phase.

A summary of the maximum area expansions achieved for foams with different alloy compositions as a function of the liquid fraction after passing the solidus temperature is presented in Figure 10. Most of the alloys studied show less expansion than the reference alloy, but some lead to a gain in maximum expansion. The latter determine the range of liquid fraction in which we can expect a good expansion. As it was shown, not only expansion is the parameter to take into account, but also foam structure, which is influenced by the amount of solid particles in the melt responsible for stabilisation. Therefore, a good compromise between high liquid fraction, facilitating expansion, and high solid fraction, enabling stabilisation, has to be found.

Finally, other process parameters, such as the foaming temperature and the temperature profile, could be further fine-tuned for each selected alloy. With all these improvements, foam quality can be positively influenced.

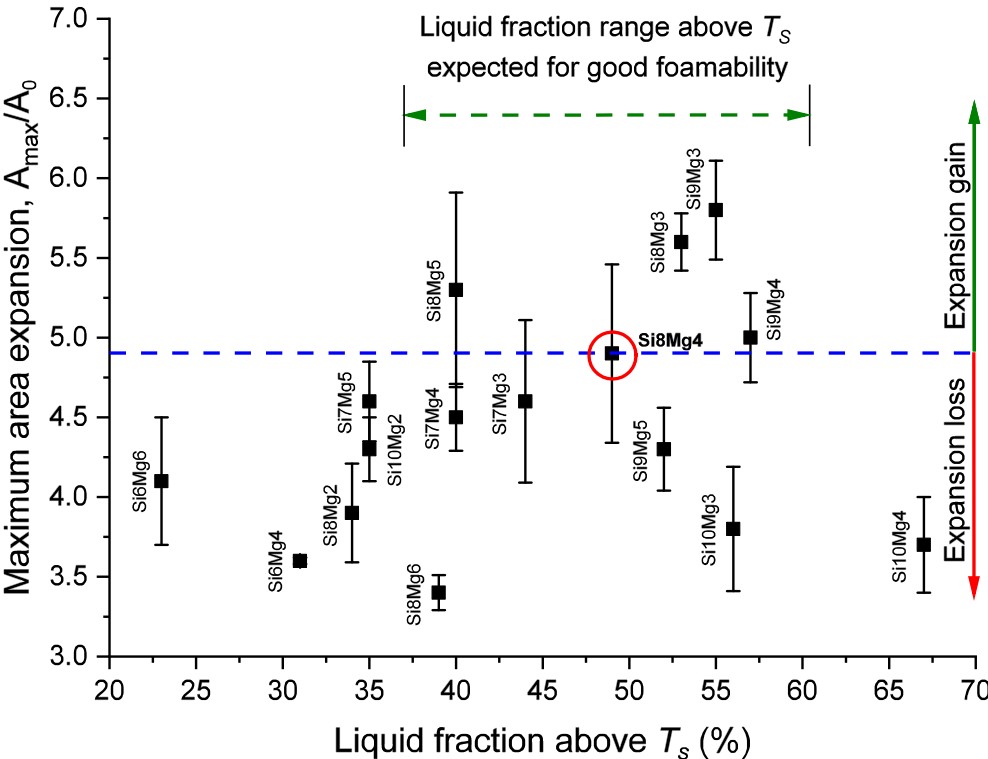

**Figure 10.** Maximum area expansion versus theoretical liquid fraction at the solidus temperature $T_S$ = 558 °C, displaying the expansion gain or loss found for different alloy compositions with respect to the reference system AlSi8Mg4. The range of expected good foamability is marked in green.

## 5. Conclusions

- An optimisation of alloy composition in the AlSi$X$Mg$Y$ system for foaming was performed for compositions around the reference AlSi8Mg4.
- A tendency to higher expansions was observed for higher Si and lower Mg contents than in the reference system.
- A correlation between increased liquid fraction and high foam expansion was found up to a preferred range for the liquid fraction of 40–60% just above $T_S$. Even higher liquid fractions had an adverse effect on foamability.
- AlSi9Mg3 shows the highest expansion of all alloys with an acceptable foam structure, while AlSi9Mg5 has the best pore structure with an acceptable expansion. Therefore 4% Mg appears as a good compromise. Within the "island of good foaming" around AlSi8Mg4 cited in the introduction, the best point might lie slightly on the Si-rich side.

**Author Contributions:** Conceptualization, F.G.-M., J.B., L.A.R. and T.R.N.; methodology, F.G.-M., L.A.R., P.H.K., T.R.N., C.M.S. and M.K.; software, P.H.K. and C.M.S.; formal analysis, L.A.R. and M.K.; writing—original draft preparation, F.G.-M.; writing—review and editing, F.G.-M. and J.B. All authors have read and agreed to the published version of the manuscript.

**Funding:** This research was funded by the Deutsche Forschungsgemeinschaft through grants GA 1304/5-1 and BA 1170/40-1.

**Acknowledgments:** We acknowledge the Paul Scherrer Institut, Villigen, Switzerland for providing synchrotron radiation beam time at the TOMCAT beamline X02DA of the SLS.

**Conflicts of Interest:** The authors declare no conflict of interest.

## Appendix A

The amount of liquid fraction above the solidus temperature of the AlSi*X*Mg*Y* system was calculated with Thermo-Calc. A variation of Si and Mg contents while keeping the other elements constant is shown in Figure A1. From these graphs, we can deduce that in this composition range, an increase of the Si content will increase the melt fraction above $T_s$, while an increase of Mg will reduce it.

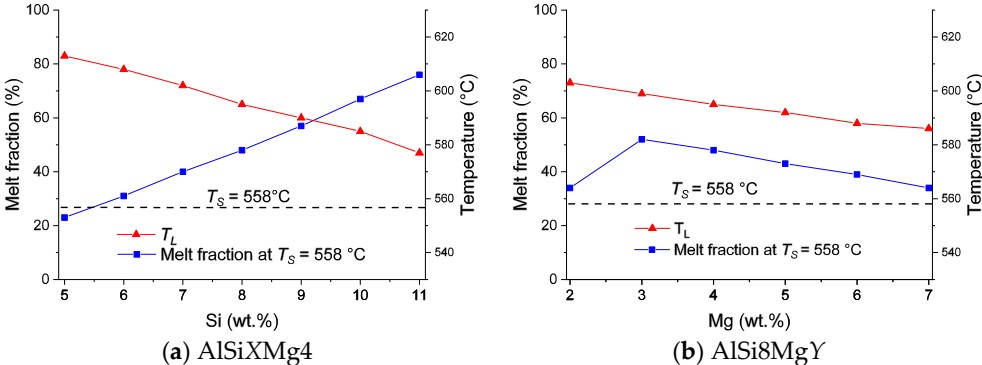

(**a**) AlSi*X*Mg4          (**b**) AlSi8Mg*Y*

**Figure A1.** Amount of liquid fraction (in blue) above the solidus temperature $T_s$ = 558 °C calculated with the software Thermo-Calc and the COST507B database for the alloys (**a**) AlSi*X*Mg4 and (**b**) AlSi8Mg*Y* for different amounts of Mg and Si, respectively. The corresponding liquidus temperatures $T_L$ are marked in red.

## Appendix B

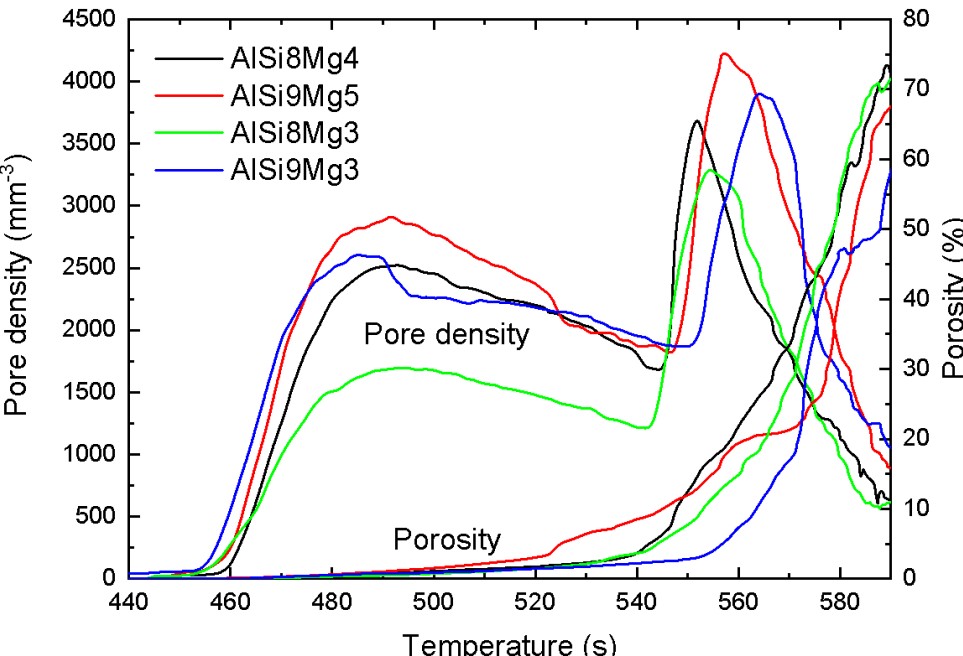

**Figure B1.** Pore density and porosity for selected samples of different alloy compositions over temperature corresponding to the samples shown in Figure 2.

## Appendix C

X-ray radiographies of the evolving liquid metal foams allowed us to follow the structure development throughout foaming. Figures C1 and C2 show the nucleation, maximum expansion, and end expansion stages for a series of alloys of composition AlSi8Mg*Y* (*Y* = 2–6 wt.%) and

AlSiXMg4 (*X* = 6–10 wt.%), respectively. Several features such as melt bubbles, cracks in early stages followed by big initial bubbles or drainage are marked in red.

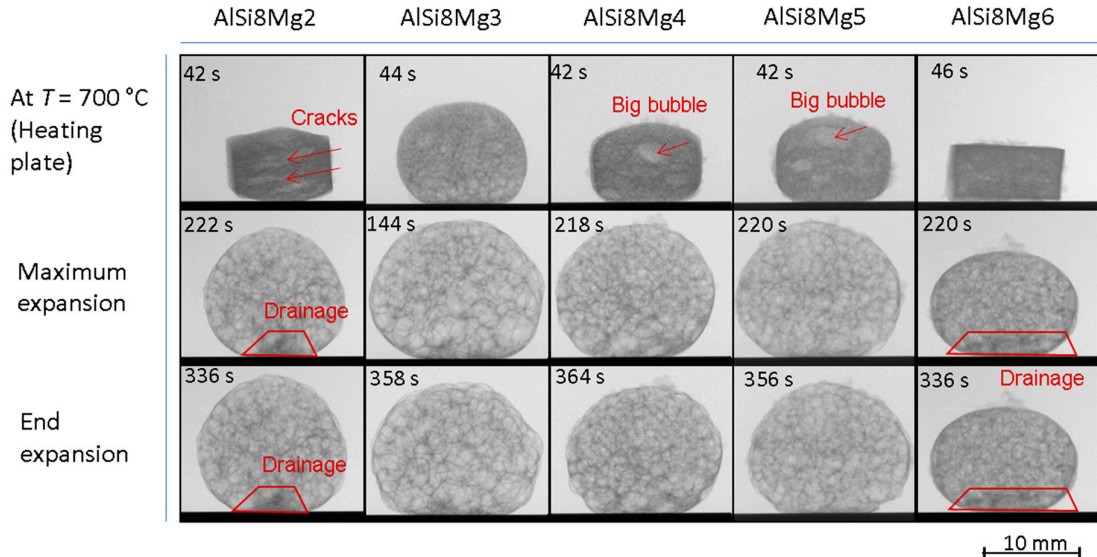

**Figure C1.** X-ray radiographies of evolving liquid metal foams of compositions AlSi8Mg*Y* (*Y* = 2–6 wt.%) extracted from an in situ radioscopic series.

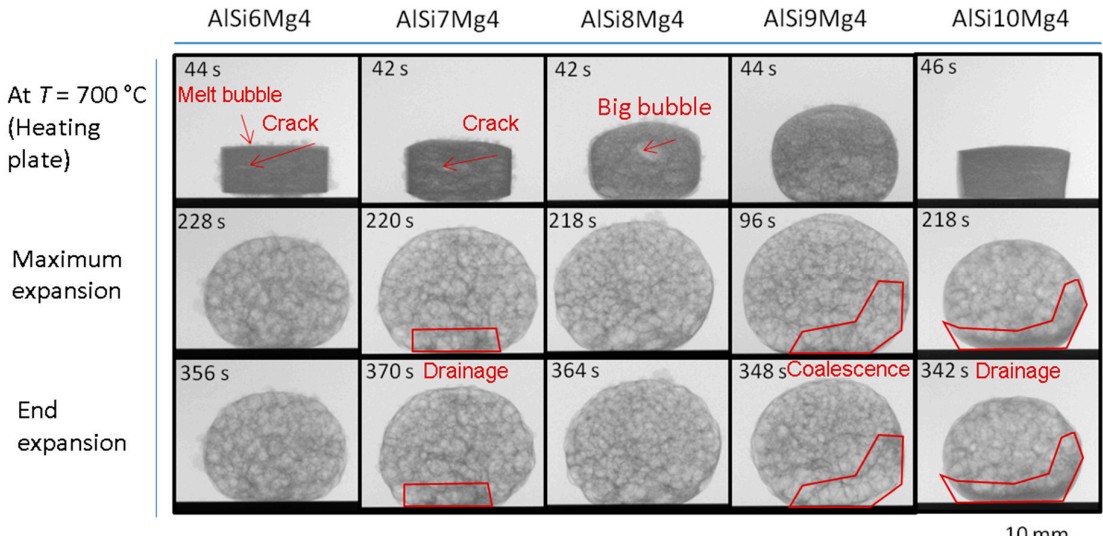

**Figure C2.** X-ray radiographies of evolving liquid metal foams of compositions AlSiXMg4 (*X* = 6–10 wt.%) extracted from an in situ radioscopic series.

**Appendix D**

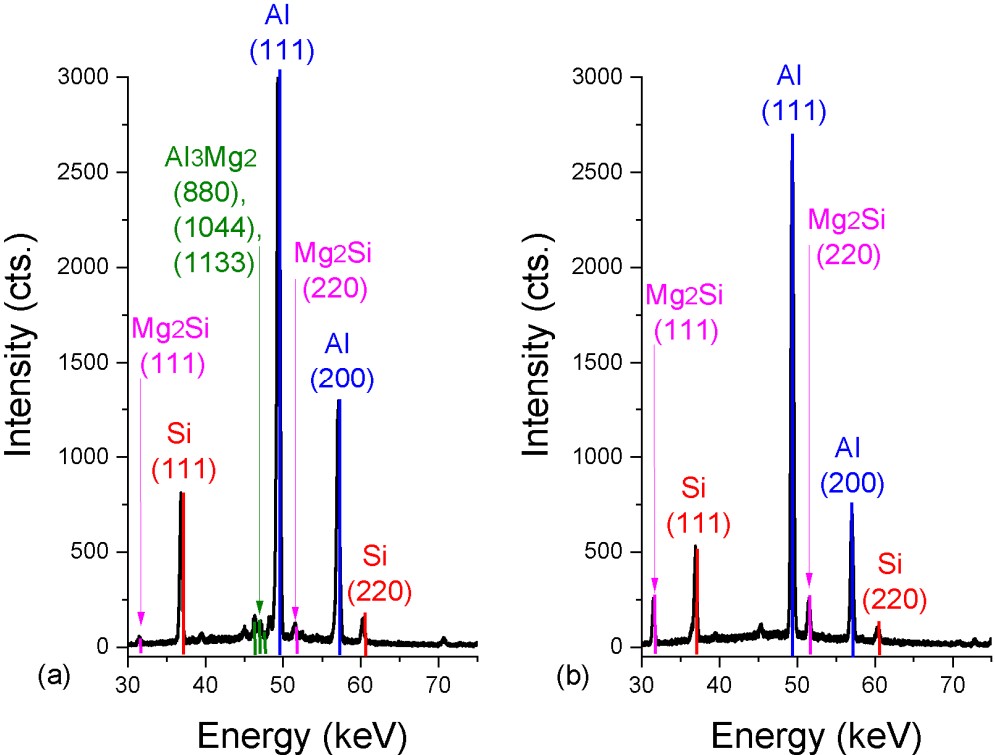

**Figure D1.** Diffraction pattern of an AlSi8Mg4 sample (**a**) before and (**b**) after foaming. The increase of the Mg2Si phase after foaming is clearly observable.

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
