# Peer review of "The Influence of Alloy Composition and Liquid Phase on Foaming of Al–Si–Mg Alloys"

_metals, doi:10.3390/met10020189_

Round 1
Reviewer 1 Report
After the reorganization, this article displays the research in a clear and solid way. The paper is fully complete and understandable to the reader.
The Authors set research goals and theses, which were implemented using appropriate research means and methods. Also, the results are at high notable level.
The research will have a significant impact on practical aspect of Al-Si-Mg alloy foam production technology, giving the technologists important information about the parameters of the production process, also explaining the foaming processes.
One remark concerns the references. The reviewer is aware of the fact that the Authors are professionals in the field of metallic foam which is evidenced by their extensive publications. Unfortunately, 25 out of 35 citations are self-citations. I suggest expanding the number of citations to some other authors in the field.
Good work Gentlemen!
Author Response
The references were updated, some old references were removed and new ones added. Now the citations/self-citations relation is 20/39.
Reviewer 2 Report
The paper topic is important and interesting from a practical point of view. This paper can be published but after the above suggestions:
Abstract: The abstract is expected to include a brief digest of the research, that is, new methods, results, concepts, and conclusions only. The abstract needs to be more focused and achievements needs mentioned clearly. At the moment abstract is more like an introduction than abstract. Please add some information from the conclusion (quantifications).
Introduction: Each one of the cited references within the body of the paper should be discussed individually and explicitly to demonstrate their significance to the study. Also note that cited authors' surname should be used as the subject of a verb, and then state in one or two sentences what they claim, what evidence they provide to support their claim, and how the work is evaluated.
The discussion is shallow and needs more details, the observations and future trends. This chapter should be connected with others published papers.
Some of the bullet points on the conclusion are simplistic; Please try to emphasize your novelty, put some quantifications, and comment on the limitations. This is a very common way to write conclusions for a learned academic journal. The conclusions should highlight the novelty and advance in understanding presented in the work.
Additionally, Authors do not write their paper in the context of Instrumentation and Measurement. Authors should present their work properly within the existing metrology literature; i.e., papers published in the metrology and measurement journals, and compare their work with these latest related Measurement work. This comparison should be done analytically (in the introduction or related work section), experimentally (in the performance evaluation section), or both.
Author Response
The main conclusion (the corroboration of the ‘island of good foaming’) is mentioned in the abstract. Some more precise quantitative results and conclusions like “a preferred range for the liquid fraction above TS between 40-60%” was added to the abstract.
Some cited authors surnames were introduced in cases where it was appropriated. The claims, results or measurements of cited authors are explained in the text.
The discussion chapter is already connected with the only one available published paper about this topic! Helwig et al. [13].
All possible quantifications are already indicated in the conclusion.
The metrology is explained and existing literature about the methods cited.
Reviewer 3 Report
line 46: What you mean as "(in wt.%)"?
Percentage of weight ratio of Si and Mg, i.e. Si 8% and Mg 4%.
You mean ... where numbers 8 and 4 defines percentage of Si and Mg components?
line 58: What you mean as "falsify"? Annul?
line 176: What dummy sample makes "dummy". It has no TiH2? Please explain idea of using such a sample. I suppose to measure internal temperature. Is dummy sample bubbling or not?
line 198: Please explain difference between maximum and end expansion. I suppose that end i the one at the end of process when specimen is solidified?
FIGURE 5: Can you somehow show this diagram also from different angle because last rows are hidden. E.g. low Mg high Si is hidden behind green bars.
Line 337: My personal comment (don't use it as relevant): If dummy specimen used for evaluation of temperature has no foaming, than probably real foaming specimen is cooler than dummy due to the expansion of gases. This is just and idea. Depends upon fact how dummy specimen is made.
FINAL COMMENT: This is really excellent research!
Author Response
Yes, this is a common way. “(where numbers define the amount of the alloying element in weight percentage, wt.%)” was added to the text for clarification.
Yes, to disprove. The text as changed accordingly.
A dummy sample is a piece of the same weight and size of a similar aluminium alloy but that does not foam to allow a good contact for the thermocouple. The text was updated accordingly.
Yes, the final expansion of the solid foam is the so-called end expansion. The text was updated accordingly.
Figure 5 was updated for a better overview.
For temperature calibration a dummy sample with the same mass and composition was used, but without TiH2. The reason is, that a foaming ample loses partially the contact to the thermocouple leading to a wrong and noisy temperature calibration. Compared to the sample mass, we believe that the expansion of the hydrogen inside the sample has not a great influence.